# BRCA Mutations and Fertility Preservation

**DOI:** 10.3390/ijms25010204

**Published:** 2023-12-22

**Authors:** Joana Dias Nunes, Isabelle Demeestere, Melody Devos

**Affiliations:** 1Research Laboratory on Human Reproduction, Université Libre de Bruxelles (ULB), 1070 Brussels, Belgium; joana.dias.nunes@ulb.be (J.D.N.); md4282@cumc.columbia.edu (M.D.); 2Fertility Clinic, HUB-Erasme Hospital, Université Libre de Bruxelles (ULB), 1070 Brussels, Belgium

**Keywords:** BRCA, fertility preservation, chemotherapy, ovarian aging, breast cancer, DNA damage

## Abstract

Hereditary cancers mostly affect the adolescent and young adult population (AYA) at reproductive age. Mutations in *BReast CAncer* (*BRCA*) genes are responsible for the majority of cases of hereditary breast and ovarian cancer. *BRCA1* and *BRCA2* act as tumor suppressor genes as they are key regulators of DNA repair through homologous recombination. Evidence of the accumulation of DNA double-strand break has been reported in aging oocytes, while *BRCA* expression decreases, leading to the hypothesis that *BRCA* mutation may impact fertility. Moreover, patients exposed to anticancer treatments are at higher risk of fertility-related issues, and *BRCA* mutations could exacerbate the treatment-induced depletion of the ovarian reserve. In this review, we summarized the functions of both genes and reported the current knowledge on the impact of *BRCA* mutations on ovarian ageing, premature ovarian insufficiency, female fertility preservation strategies and insights about male infertility. Altogether, this review provides relevant up-to-date information on the impact of *BRCA1/2* mutations on fertility. Notably, *BRCA*-mutated patients should be adequately counselled for fertility preservation strategies, considering their higher sensitivity to chemotherapy gonadotoxic effects.

## 1. Introduction

In 2020, breast cancer (BC) was diagnosed in more than 2.26 million women worldwide. Female BC represents around 12% of all cancer diagnosed among both sexes, making it the most common cancer worldwide [1]. In Europe, it is estimated that 9% of women will develop BC in their lifetime [2,3]. Although the median age at diagnosis is 62 years, around 5–10% of BC cases occur in women under 39 years old [4,5]. For men, less than 1% of BC cases are diagnosed in this population [6]. Breast cancer can be classified according to its molecular subtype, and hormonal receptors status: estrogen receptor positive (ER+) and/or progesterone receptor positive (PR+); Human Epidermal Growth Factor Receptor-2 (HER2) positive (HER2+); or triple negative (ER−, PR−, HER2−) breast cancer (TNBC) [7].

Around 10% of BC is attributable to inherited genetic predisposition affecting mostly adolescent and young adults (AYAs). Several germline mutations that increase the risk of developing BC have been identified in genes such as *BReast CAncer* genes *1* and *2* (*BRCA1/2*), *Tumor protein p53* (*TP53*), *Partner and Localizer of BRCA2* (*PALB2*), *phosphatase and tensin homolog* (*PTEN*) and *mutated ataxia-telangiectasia* (*ATM*) [8]. The most prevalent mutations occur in *BRCA1* and *BRCA2* genes with around 50% of young hereditary BC patients carrying a germline mutation for one of those genes [9,10]. In addition to BC, both gene mutations expose carriers to an increased risk for ovarian, pancreatic and prostate cancers [11]. Cancer predisposition varies according to the mutated gene and the sex of the carrier. The risk of developing BC reaches 85% and 40–45% for *BRCA1* and *BRCA2* female mutation carriers, respectively. For ovarian cancer (OC), the risk is around 40–50% for *BRCA1* and 15–30% for *BRCA2* mutation carriers [12]. For men, *BRCA1/2* mutation carriers face a higher risk of developing BC in comparison to the general population, while *BRCA2* mutation carriers have an increased risk of developing prostate cancer [11].

Although carrying a *BRCA1/2* germline mutation significantly increases the risk of developing cancer, their prevalence in the general population is low (0.1–0.2%) [12]. Nevertheless, since the AYAs cancer patients are more likely to carry one of these germline mutations, the improvement of our knowledge on the impact of these mutations on their future fertility, the ovarian response to oncological treatment and the fertility preservation outcomes became a research priority during the last decade.

## 2. BRCA

### 2.1. Heritance of Breast Cancer

Reports of families affected with breast cancer over generations have been documented since the mid-19th century [13]. In the 1980s, Mary-Claire King’s team proposed a model of BC heritance. Thanks to segregation analysis in 1579 families affected with BC, they were able to link BC to an autosomal dominant high penetrant susceptibility gene. Women carrying this allele were at 82% risk of developing BC [14]. A few years later, this gene was mapped onto chromosome 17q21 by studying genetic markers from 23 families with early-onset familial breast cancer [15]. In 1991, this susceptible gene was named *BRCA1* and its sequence was later published by the Myriad group [16,17]. The same year, another susceptibility gene was localized on chromosome 13q12-13, sequenced and named *BRCA2* [18,19]. For almost 20 years, the Myriad group had a patent on *BRCA1* and *BRCA2* testing, which was ruled out in 2013, allowing broader availability of genetic testing among individuals at risk [13,16]. Notably, those germline mutations were more commonly detected in Ashkenazi Jews population, among whom pathogenic variants are present in approximately 2.5% of the individuals, of which 185delAG and 5382insC in *BRCA1*, and 6174delT in *BRCA2* [20].

### 2.2. BRCA Genes Structure

*BRCA1* and *BRCA2* genes act as tumor suppressor genes. While both are involved in DNA repair mechanisms, their structure and function differ from each other (Figure 1).

*BRCA1* gene contains 43,044,294 to 43,125,482 base pairs (bp) with 24 exons. The translated protein of 1863 amino acids (aa) is composed of three domains: the Really Interesting New Gene (RING) domain (exons 2–7; aa 1–109), a region encoded by exons 11–13, and the BRCA1 C-Terminus (BRCT) domain (exons 16–24; aa 1650–1863) [21,22]. The RING domain contains a zinc-binding RING finger motif that ensures the E3-ubiquitin ligase activity of BRCA1 once it is bound to BRCA1-associated RING domain protein 1’s (BARD1) RING finger. This interaction covers the C-terminal nuclear export sequences (NES) located in the RING domain of both proteins, allowing the nuclear retention of BRCA1/BARD1 heterodimer [23]. The region encoded by exons 11–13 covers up to 65% of *BRCA1* and contains protein binding sites for DNA repair proteins RAD50 and RAD51, Retinoblastoma (Rb) and cellular Myelocytomatosis (c-Myc) [23,24]. It is composed of two nuclear localization sequences (NLS), a coiled-coil domain interacting with the PALB2 and a serine cluster domain (SCD). The C-terminal region is constituted of two connected BRCT domains allowing for the recognition of phosphorylated serine-X-X-phenylalanine (S-X-X-F) motifs on partner proteins involved in DNA repair such as C-terminal binding protein 1 Interacting Protein (CtIP), BRCA1 A Complex Subunit (ABRAXAS), BTB domain and CNC homolog 1 (BACH1), BRCA1 interacting protein C-terminal helicase 1 (BRIP1) and p53. The macromolecular complexes formed allow the selection of the substrates for BRCA1/BARD1 E3 ubiquitin ligase activity [22,23].

The *BRCA2* gene contains 32,315,479 to 32,399,671 base pairs with 27 exons. It encodes for an 3418 amino acids protein deprived of enzymatic activity and divided into three main regions of interest: the N-terminal Transcriptional Activation Domain (TAD) (exons 2–3; aa 15–105), eight BRC motifs (exon 11; aa 1002–2085) and one DNA Binding Domain (DBD) in the C-terminal region (exons 12–27; aa 2482–3184) [21,22]. The N-terminal region is composed of a TAD, allowing PALB2/BRCA2 interaction [22,25], and phosphorylation sites available for cyclin-dependent kinase 1 (CDK1) at Ser93, Thr64 and 77 and for polo-like kinase 1 (PLK1) at Ser193, 205, 206, Thr203 and 207 [26,27]. Exon 11 is the largest region of BRCA2 and contains eight BRC repeats. Each conserved motif is 35 aa long and allows the binding to monomeric RAD51. In the C-term, the DBD is composed of one helical domain (HD), three oligonucleotide-binding (OB) folds. This rearrangement allows BRCA2 to bind to single and double strand DNA [25,27]. Moreover, the BRCA2 C-terminus contains three NLS and a C-terminal RAD51 interaction domain (CTRD), previously referred to as the TR2 domain, stabilizing the BRCA2-RAD51 interaction. Phosphorylation sites are also present at the C-term for CDKs and checkpoint kinases 1 and 2 (CHK1/2) at Ser 3284, 3291, 3319, Thr 3310 and 3323, and at Ser 3387, respectively [26,28].

### 2.3. BRCA and Genome Integrity

Throughout their lifespan, cells are subjected to several damages to their genome. Cells need to rely on deep regulated mechanisms to scan and repair the DNA damages as their identity and viability depends on genome integrity [29]. DNA double strand breaks (DSBs) may be caused by exogenous agents (as irradiation and chemical agents such as chemotherapy) or endogenous processes (DNA replication and repair) [30,31]. Several mechanisms ensure DNA repair, among them, the homologous recombination (HR) and non-homologous end joining (NHEJ) [32,33,34]. HR and NHEJ are activated depending on the cell cycle and more specifically during S/G2 and G1 phases, respectively. Opposite to HR, NHEJ is an error-prone DNA repair pathway and may lead to chromosome rearrangement and hence to genomic instability. DSBs are recognized by sensors and a DNA damage response (DDR) is triggered through the transmission of the signal to effectors, which initiate repair by mediators [35]. Despite their structural and functional differences, both BRCA1 and BRCA2 are involved in DNA repair through HR.

DSBs are detected by ATM and MRN complex, which is composed of the DNA repair proteins MRE11 and RAD50, and Nibrin (NBS1) [36]. ATM activation is mediated through the recruitment of the MRN at the DSBs site, autophosphorylation and acetylation [37]. ATM then phosphorylates the H2A histone family member X (H2AX) to trigger the reparation machinery and the Mediator of DNA damage Checkpoint Protein 1 (MDC1) to amplify the DDR [38]. Follows the recruitment of E3-ubiquitine ligases, such as RING Finger proteins 8 and 168 (RNF8/168), which leads to a ubiquitin signaling cascade on phosphorylated H2AX (γ-H2AX) [39,40]. This signal recruits the tumor suppressor p53-binding protein 1 (53BP1) and BRCA1 to the DSB site by MDC1 and receptor-associated protein 80 (RAP80), respectively. Depending on the cell cycle, a switch occurs between 53BP1 and BRCA1 localization at DSBs site. Cells in the G1 phase will be repaired by NHEJ due to the recruitment of 53BP1 after histone modifications, of which H2AX phosphorylation. However, in the S phase, CtIP is phosphorylated by CDK, which induces its interaction with BRCA1 and the MRN complex. The formation of this complex, named the BRCA1-C complex, triggers the removal of 53BP1 at DSB sites and the repair of DNA DSBs by HR [41]. BRCA1 is phosphorylated during the S phase by ATM/ATR and CHK1/2 to allow its nuclear import to DNA DSBs. BRCA1 forms a macrocomplex (BRCA1-A complex) with ABRAXAS and RAP80, to maintain DDR signaling and the activation of the G2/M checkpoint [35,42]. The BRCA1-C complex also initiates the end resection of DNA by generating single strand DNA (ssDNA) [43]. Whilst end resection does not depend on BRCA1, the protein facilitates the process [42]. Extensive end resection is supported by exonuclease 1 (EXO1) and DNA2 to allow Replication Protein A (RPA) loading on ssDNA. Once phosphorylated by CHK2, BRCA1 forms a complex with PALB2/BRCA2 to replace RPA with RAD51. DNA damage also triggers BRCA2 phosphorylation by CHK1/2 to enhance its interaction with RAD51. Besides its RAD51 binding sites, BRCA2 binds to DSS1, a protein that mimics ssDNA, so RPA will bind to DSS1 and RAD51 will be loaded on free ssDNA [43] (Figure 2). DNA repair by HR is completed once strand invasion on the homolog chromatid occurs.

In addition to its known functions in HR, BRCA1 is part of a BRCA1-associated genome surveillance complex (BASC), composed by MRN complex, ATM, DNA mismatch repair proteins 1 (MLH1), 2 and 6 (MSH2-6), bloom syndrome protein (BLM) and DNA replication factor C. However, despite a potential role as DNA damage sensor, the function of this complex remains elusive [44].

Besides DNA repair mechanisms, BRCA1 also plays a role in cell cycle checkpoint activation. Its interaction with BARD1 promotes the phosphorylation of p53 by ATM, leading to the transcription of p21 and cell cycle arrest in the G1 phase. Through its interaction with BRIP1 (also known as FANCJ) [45,46] and DNA topoisomerase 2-binding protein 1 (TOPBP1), BRCA1 facilitates CHK1 phosphorylation by ATM. Phosphorylated CHK1 activates WEE1 and inhibits M-phase inducer phosphatase 1–3 (CDC25A-C). This leads to the inhibition of CDK1/2, hence cell cycle arrest at S phase and G2/M phase [35]. BRCA1 also inhibits directly proliferation and cell growth through interactions with Rb protein and the inhibition of c-Myc activity [24]. BRCA1 also acts on chromatin remodeling through its interaction with SWItch/Sucrose Non-Fermentable subfamily (SWI/SNF). BRCA1 function is not limited to the nucleus, as an apoptotic pathway dependent on BRCA1 and its inhibition mediated by BRCA1/BARD1 interaction has been reported. Finally, BRCA1’s interaction with BARD1 and Obg-like ATPase 1 (OLA1) regulates the number of centrosomes and its implication in mitophagy remains to be elucidated [47]. It has also been demonstrated that BRCA1 plays a role in cancer stem cells’ (CSC) development and evolution by regulating several signaling pathways such as phosphatidylinositol-4,5-bisphosphate-3-kinase (PI3K)/protein kinase B (AKT), Hedgehog, Janus kinase (JAK)—Signal Transducer and Activator of Transcription proteins (STAT) and Notch [21].

BRCA2 is mainly involved in DNA repair through HR but is also involved in intra-strand crosslinks (ICLs) [46] and programmed DSBs during meiosis. Moreover, it has also been shown that BRCA2 is involved in the protection of telomere integrity and stalled replication fork degradation. Indeed, BRCA2 prevents the end resection action of nucleases, such as MRE11, by stabilizing RAD51 filaments [21,35]. Finally, BRCA2 is thought to be involved in mitosis following phosphorylation by CHK1/2 and PLK1. During the metaphase to anaphase transition, the attachment of duplicated chromosomes to the spindle relies on the spindle assembly checkpoint (SAC) [48]. This complex is composed, among other proteins, of budding uninhibited by benzimidazole-related 1 (BUBR1) protein which needs to be acetylated to interact with the anaphase promoting complex (APC/C). BRCA2 acts as a scaffold by bringing together BUBR1 and its acetyltransferase p300/cAMP response element-binding protein (P/CAF). BRCA2 also interacts with proteins involved in cytokinesis. By binding to Filamin A, BRCA2 is recruited at the midbody and interacts with a centrosomal protein of 55 kDa (CEP55). This interaction will induce the recruitment of endosomal sorting complexes required for transport (ESCRT)-associated proteins, allowing the cleavage of the membrane bridge [26].

### 2.4. Pathogenic Variants

*BRCA* mutations are continuously reported in online databases such as the Breast cancer Information Core (BIC), the BRCA Exchange and ClinVar. These platforms allow experts to upload the variants of (un)known significance they encounter but also to assess the pathogenicity of the variant their patient harbors. On the BRCA Exchange website, around 7% of the reported *BRCA* variants are pathogenic. The three most common mutations can be found among these pathogenic variants: 185delAG (*BRCA1*), 5382insC (*BRCA1*) and 6174delT (*BRCA2*) [21,49].

Breast and ovarian cancer cluster regions (BCCR and OCCR, respectively) have been mapped on both *BRCA1* and *BRCA2* genes (Figure 1). This classification identifies in which region a mutation is more likely to encode a *BRCA* pathogenic variant with an increased risk of BC and decreased risk of OC within BCCR, and an increased risk of OC and decreased risk of BC in OCCR. *BRCA1* has two BCCR and one OCCR distributed along its sequence. BCCRs can be found on the N- and C-terminal regions (BCCR1: c.179–505; BCCR2: c.4328–4945 and BCCR2′: c.5261–5563), corresponding to the RING and BRCT domains, respectively. Whilst 5382insC can be found within BCCR2 (c.5266), 185delAG is located right before BCCR1 (c.68_69), meaning that the risk for BC and OC is equivalent. OCCR is located along exon 11, from c.1380 to c.4062. *BRCA2* contains two BCCRs and two OCCRs. BCCR1 is distributed along the N-terminal region of *BRCA2* (BCCR1: c.1–596 and BCCR1′: c.772–1806) and BCCR2 within the DBD (c.7394–8904). On the other hand, OCCRs are found in the central region, with OCCR1 located on exon 11 within the BRC repeats (c.3249–5681 and c.5946) and OCCR2 within c.6645–7471. The third founder mutation, found within *BRCA2*, is located on OCCR1 (c5946), suggesting a higher risk of developing OC than BC when carrying this mutation [50]. Interestingly, male carriers of pathogenic *BRCA2* mutation variants have a lower risk of developing prostate cancer (PCa) when the variant is located within *BRCA2* OCCR1. In fact, men carriers of the *BRCA2* founder variant 6174delT have lower risk of developing PCa than carriers of other *BRCA2* mutations [51]. Instead, a prostate cancer cluster region (PCCR) was identified in *BRCA2* gene from c.7914 to 3′, but no PCCR was established in *BRCA1* [52].

## 3. BRCA Mutations and Fertility

### 3.1. Ovarian Ageing

Women’s reproductive potential is ensured by the follicles and functional units present at different stages of development within the ovary. Unlike men, women have a limited stock of gametes determined at birth [53]. During female fetal life, primary germ cells differentiate into oogonium, and by week 20, around 7 million of them have been produced. Along this process, oogonium are blocked in first meiosis at prophase I, which marks the end of the production of new germ cells. Simultaneously, oogonia differentiates into primordial follicle (PF), an oocyte surrounded by flattened granulosa cells in a quiescent state. At birth, only 2 million PFs are present due to follicle apoptosis-mediated atresia. From birth to menopause, the primordial follicle pool, constituting the ovarian reserve, declines, reaching 400,000 PFs at puberty and less than 1000 PFs at menopause [54]. Through time, PFs will face different paths: (1) remaining in the quiescent stage or (2) entering the growing phase, growing follicles will then (3) undergo atresia or (4) develop into a dominant follicle which will resume meiosis and be ovulated. Ultimately, the vast majority of follicles will undergo atresia as less than 1% will achieve ovulation [55]. Although older models indicated that follicle depletion accelerated abruptly around 37 years, current models assess that the number of PFs decreases constantly over time [56]. However, fertility changes are clinically observed from the mid to late 30s [57]. Circulating anti-Müllerian hormone (AMH) is a common clinical parameter used to evaluate the ovarian pool. Produced by the granulosa cells of growing follicles, this hormone inhibits the activation of PFs, hence maintaining their quiescent state. With the diminution of the ovarian reserve over time, and the subsequent PFs recruitment, the level of AMH naturally decreases [58].

Ovarian ageing is defined as the quantitative and qualitative oocyte decline throughout a woman’s reproductive lifespan. Menopause is a physiological event that occurs around 45 and 55 years [59]. With the diminution of the ovarian reserve, hormonal levels of inhibin-B and AMH decrease. In response to this process, follicle stimulating hormone (FSH) and estradiol levels increase briefly before estradiol collapses, inducing clinical symptoms [60]. Menopause is clinically characterized by the cessation of the menstrual cycles for more than 12 months, circulating FSH levels above 25.8 IU/L associated with AMH level below detectable levels [61,62]. While the impact of lifestyle on ageing is still debated [56], few factors have been identified as being regulators of ovarian aging (reviewed in Park S. et al., 2021 [63]). Some of the features highlighted involve genomic abnormalities such as chromosome mis-segregation, recombination errors during meiosis or diminished activity in the SAC, all contributing to aneuploidy [63]. DNA-related factors include the diminished expression of genes involved in DNA repair (discussed further subsequently), telomere shortening and the accumulation of mitochondrial DNA damages due to reactive oxygen species (ROS). At last, genetic factors including mutations in genes involved in DNA repair pathways—like the Fanconi anemia (FA)/BRCA pathway, active during the repair of ICLs—or mutations in genes associated with premature ovarian insufficiency (POI)—of which Fragile X Messenger Ribonucleoprotein 1 (FMR1)—can lead to premature ovarian ageing [63,64].

Within the oocyte, DNA DSBs occurred when undergoing meiosis. DSBs can be induced either in the fetal ovary, when meiosis I is initiated, or in the post-natal ovary, when meiosis II occurs, resulting in the expulsion of the second polar body. However, DSBs can also be unprogrammed due to exogeneous factors [65]. When DNA DSBs occur in quiescent PFs, HR is believed to be the main pathway recruited for DNA repair [66]. Meiosis arrest holds PFs in the G2/M cell cycle phase, promoting DNA damage to be handled by ATM-related DNA repair pathway. Indeed, sister chromatids are already present, and oocytes must be repaired by error-free mechanisms to avoid germline mutations, otherwise follicles undergo apoptosis [65,66]. DNA DSBs events accumulate physiologically with aging and NHEJ seems to participate in the repair of DNA damages occurring in mature oocytes [67]. Several studies have assessed the gene expression profile of young and aged follicles or oocytes, and underscored a decrease in DNA repair genes expression with age in rodent [68,69,70,71,72,73,74], cattle [75], monkey [76] and human [77,78,79,80] models. Notably, reduced expression of *Brca1* was observed in metaphase II (MII)-arrested eggs from old mice compared to younger ones [81]. Similarly, a decrease in BRCA1 expression and its activated state, p-BRCA1, was observed with aging in PFs isolated from rats [69], as well as in buffalos’ ovaries [82]. Human studies reported a similar decrease in the expression of *BRCA1* and the accumulation of DNA DSBs in aged oocytes compared to young ones [68]. Therefore, during their lifespan, DNA damages accumulate in the oocyte, and in contrast, the expression of genes involved in DNA repair mechanisms decreases [68]. Altogether, those events lead to higher apoptosis levels in aged oocytes.

A human comparative study assessing the impact of *BRCA*-mutation on DNA damage level reported a higher ratio of γ-H2AX in *BRCA1*-mutated PFs, and highlighted that DNA DSBs accumulation is accelerated in *BRCA*-mutated individuals over 30 years [83]. These observations suggest that *BRCA* mutations may accelerate ovarian ageing by impacting oocyte quality, leading to premature exhaustion of the ovarian reserve. However, the effect of *BRCA* mutations on the onset of menopause is not clear with divergent clinical studies, from lower to normal age of natural menopause for *BRCA*-mutated patients [84,85].

### 3.2. Premature Ovarian Insufficiency

Menopause can occur in younger women and is defined as premature ovarian insufficiency (POI), a disorder affecting 1% of the general population [86]. Clinically, POI affects women under 40 years old and is described as a hypergonadotropic hypogonadism state along with 12 months of oligo/amenorrhea, high gonadotropin (FSH > 25 IU/L repeated at four weeks interval) and low estradiol levels (<50 pg/mL) [87,88]. Although POI etiology is idiopathic in 90% of cases, the most commonly identified causes include genetic factors, oncological treatments or infections. The effect of chemotherapy/radiotherapy has been well studied and points to a diminished ovarian reserve after exposure [89]. By targeting both the oocyte and the granulosa cells, those treatments induce the apoptosis of PFs and growing follicles, but also the activation of PFs, diminishing the ovarian reserve by a process called the “burn-out effect” [90]. Chemotherapy-induced gonadotoxicity varies highly according to the therapeutic schema. Nevertheless, most POI cases reported were due to alkylating and platinum-based agents, taxanes, anthracyclines, topoisomerase inhibitors and vinca alkaloids [91].

In the past few years, studies have been pointing out the potential impact of *BRCA* mutations on fertility and on the risk of developing POI. Indeed, a decreased ovarian pool and increased fertility-related issues have been observed in *BRCA*-mutation carriers [92]. Although no impact of *BRCA* mutations was reported on nulliparous rate, concerns were raised in assisted reproductive technologies (ART) field regarding its effect on ovarian stimulation (OS) response [84]. Whilst controversies remain regarding the impact on the total oocytes retrieved after gonadotropins stimulation, several studies observed a decreased number of mature oocytes in the *BRCA*-positive population (reviewed in [84,93]). Interestingly, it appeared that the number of oocytes retrieved was lower in *BRCA1*-mutated patients compared to *BRCA2*-mutated ones [94].

Several studies have compared the level of AMH between *BRCA*-mutated and non-mutated individuals (Table 1). Out of the ten studies reviewed in Hu et al. (2020), one had a statistical decrease in AMH in *BRCA*-mutated patients [68], one showed a statistical increase in AMH in the mutated cohort [95] and the other studies had no statistical differences between the two populations. Similarly, a meta-analysis on six studies suggested no effect of *BRCA* genes mutations on AMH levels [84]. More recently, a meta-analysis performed on five data sets including 824 women, showed significantly lower AMH levels in women harboring a *BRCA* germline mutation, and more specifically *BRCA1* mutations, compared to controls [96]. Those findings are in accordance with another report based on 308 women [94]. Another adjusted meta-analysis reached the same conclusion in a population under 42 [97]. However, this effect was not observed in three other studies (two retrospectives and one prospective) from South Korea, The Netherlands and Belgium [98,99,100].

Few studies have assessed how follicle density is affected by *BRCA* mutations within ovarian tissue harvested for fertility preservation (Table 1). In a study involving premenopausal patients who had undergo oophorectomy, follicle counting revealed that *BRCA*-mutated patients had a significantly lower number of follicles compared to controls [109]. Specifically, PFs density tends to be lower in *BRCA*-mutated patients’ tissue compared to control ones, independently of the type of gene. Whilst non-significative, another study reported a lower number of oocytes in *BRCA*-positive patients compared to *BRCA*-negative patients [104]. Clinical antral follicle count (AFC), performed through transvaginal ultrasound, showed more controversial results [110]. While one study observed higher AFC in *BRCA*-mutated carriers compared to non-carriers [106], two other studies did not observe any differences between the two populations [99,105].

### 3.3. Female Fertility Preservation

In the past few years, due to the increasing survival rates observed in cancer patients, improving the quality of life for cancer survivors became a priority. All cancers combined, 5-year survival rates reached 84% and 86.7% in prepubertal and AYA patients, respectively [111]. One of the major issues encountered by young survivors is fertility-related impairment induced by oncological treatment [112].

While the impact of some of the most commonly administrated drugs, such as cyclophosphamide, cisplatin and doxorubicin, on fertility have been established, the gonadotoxic effect of newer drugs, such as poly-adipose polymerase inhibitors (PARPi) or drug combinations, is unknown [113]. The widespread use of PARPi in the regimen of *BRCA*-mutated patients is based on the principle of synthetic lethality. Due to the inability of mutation carriers to efficiently repair DNA DSBs, PARPis, such as olaparib, inhibit the repair of the DNA single strand break created by other toxic compounds, which enhances apoptosis of the cell. Studies in mice reported the gonadotoxicity of olaparib [114,115]. However, no reports on the toxicity of olaparib on human ovarian tissue are available. Further fertility-related studies are needed regarding chemotherapy-induced gonadotoxicity in *BRCA*-mutated patients due to their vulnerable reproductive potential. Nevertheless, a consortium of experts has published specific guidelines regarding the established gonadotoxicity of anti-cancer treatments [113,116]. It is essential to offer counselling for fertility preservation strategies to these patients to ensure an optimal quality of life after the treatment. Different strategies are available in clinics, such as oocyte/embryo or ovarian tissue cryopreservation. Oocyte and embryo cryopreservation were established in the 1980’s and are considered as the gold-standard procedure for women willing to preserve their fertility [117]. The first livebirths after oocyte or embryo cryopreservation were reported shortly after its discovery [118,119]. Oocyte vitrification and embryo cryopreservation have the most successful live birth rates (LBR) among fertility preservation strategies, reaching 32% and 41%, respectively [120]. Due to the COS procedure, these methods are not available for prepubertal patients nor for patients who need to start emergency chemotherapy or radiotherapy. OTC is the only suitable procedure for prepubertal patients; it does not delay the start of anticancer treatments and enables the return of the endocrine function of the ovary. Cortical ovarian stripes are surgically collected from the patient and cryopreserved until cancer remission. Following grafting, spontaneous LBR in this procedure reaches 33% [120]. Limitations of OTC include the risk of malignant cells reseeding, invasive surgery, risk of ischemia-perfusion and graft survival [121]. Several studies supported the safety of ART for *BRCA* mutation carriers, before or after BC. Nevertheless, carrying this mutation can impact the outcome and success rate of such procedures [122,123].

*BRCA*-mutated patients have fewer mature oocytes following COS and higher poor response rates than non-carriers [84,93]. In addition, the cryopreservation of oocytes/embryos at a younger age may be favorable as the depletion of their ovarian reserve is suspected to be accelerated [124]. Yet, these techniques remain suitable for *BRCA*-mutated patients [92,122,125]. Initially, COS was not recommended for patients with estrogen-sensitive cancers, but it is now performed with the introduction of aromatase inhibitors in the stimulation protocol used for fertility preservation at diagnosis [126,127]. To prevent a gonadotropins-induced peak in estradiol, aromatase inhibitors such as letrozole are administered concomitantly to inhibit the conversion of androgens to estrogens, limiting estrogen concentration in plasma. In a study based on the co-administration of letrozole and recombinant FSH (LF), the number of total and matured oocytes, as well as the number of cryopreserved embryos, was higher in patients treated with LF than patients not treated with letrozole [128]. However, a meta-analysis based on the results from eleven studies showed no difference in the number of MII or total oocytes, neither in the maturation rate, in letrozole versus non-letrozole cohorts [127]. In addition to its inhibition of estrogen production, letrozole induces an increase in androgen concentrations, which in turn improves OS response. Whilst the results on OS response following the use of letrozole vary between studies on *BRCA*-mutated patients, the use of tamoxifen as an aromatase inhibitor did not show any difference between mutated and non-mutated patients [125].

*BRCA*-mutated carriers have increased risk of developing OC, so they are offered risk-reducing surgery in the form of bilateral salpingo-oophorectomy (BSO) from the age 35 or 40 when harboring germline *BRCA1* or *BRCA2* mutations, respectively [129]. Thus, fertility preservation can be offered to young BRCA mutation carriers to prevent infertility risk, especially for those who would like to benefit from preimplantation diagnosis or can be still offered to young *BRCA*-mutation cancer patients if the treatment needs to be started urgently.

In the case of ovarian tissue cryopreservation, the tissue should be transplanted to the site of the remaining ovaries to easily remove it during the prophylactic BSO surgery after the completion of their reproductive plans. As the ovarian pool is suspected to be lower in *BRCA*-mutated patients, fragments may contain fewer follicles, which could impact the LBR after grafting. Nevertheless, several pregnancies have already been reported in *BRCA*-mutated cancer survivors following this procedure. Although it is not the first option to offer to these patients, this fertility preservation strategy seems safe and effective [122,125].

### 3.4. BRCA Mutations and Male Infertility

Although less common, male carriers of *BRCA1/2* mutations are also at higher risk of developing cancers compared to the unaffected population. Unlike women, male cancer predisposition arises when harboring mutations in the *BRCA2* gene. Whilst accounting for a small proportion of cancer cases—0.1% of BC and around 1% of PCa—the risk for carriers to develop BC increases up to 1% and 7–8%, while for PCa it ranges to almost 4-fold increase and from 3 to 8.6-fold increase, for *BRCA1* and *BRCA2* mutations, respectively [130,131,132]. Studies focusing on fertility-related issues in male carriers of *BRCA* mutations are scarce. In a *Palb2*-deficient mouse model, mutant males harbored a defect in spermatogenesis with increased germ cell apoptosis and produced smaller litters compared to wild-type (WT). These results highlighted the importance of BRCA1-PALB2-BRCA2 in DNA repair and male gametes meiosis [133]. The link between *BRCA* genes’ regulation and male infertility was further assessed in human model using infertile patients’ semen. However, no clear correlation between the methylation level of *BRCA1* and *BRCA2* promoters and sperm DNA fragmentation was reported [134]. A small cohort study assessed the impact of the mutation on hormone levels and reported increasing levels of total testosterone and free androgen index in *BRCA* mutation carriers [135]. However, a recent study on a larger cohort of men at increased genetic risk of PCa did not show any difference in testosterone levels (total and free) nor in sex hormone binding globulin levels in *BRCA*-mutated carriers compared to non-carriers [136].

Similar to women, male cancer patients expose to chemotherapy are at risk regarding their fertility [137]. Platinum-based therapy is commonly offered to *BRCA*-mutated carriers. Cisplatin and carboplatin are two platinum-derived drugs, and whilst carboplatin is considered to be less toxic than cisplatin, only a few data are available regarding fertility. In an in vitro culture and xenograft model study exposing prepubertal testicular tissue to cisplatin or carboplatin, the effect of chemotherapy-induced germ cell loss was similar for both drugs [138]. As for women, consortium established fertility preservation guidelines are continuously updated. Sperm freezing is the standard technique for fertility preservation of post-pubertal patients. Testis tissue cryopreservation is the only available solution for prepubertal patients; however, it is still an experimental procedure [139].

## 4. Perspectives and Research Priorities

Despite differing in their structure and their role, *BRCA1* and *BRCA2* are both key players in DNA DSBs repair through HR. They act as tumor suppressor genes and their mutations predispose patients to breast, ovarian, prostate and pancreatic cancers. It was suggested that their high DNA repetitive elements content was responsible for their high genomic instability. Although ubiquitously expressed, the reason cancer develops in a tissue-specific manner in *BRCA1/2* mutation carriers is not fully understood. Studies on breast and ovarian cells revealed that *BRCA*-deficient cells develop tumor instead of undergoing apoptosis [140]. However, as a deficiency in *Brca* genes causes mouse embryonic lethality [141,142], it is challenging to explore the tumorigenesis arising from the loss of function of those genes in vivo. A few studies hypothesized that the tissue-specificity is due to estrogen expression. Indeed, breast and ovaries are the main organs affected by *BRCA* mutations and both are regulated by sex hormones [143]. Breast epithelial cells start proliferating at puberty in response to estrogen production from the ovaries. Interestingly, the expression of *BRCA* genes is upregulated during puberty and pregnancy, hence the suggestion that *BRCA* expression is stimulated by estrogen production [140]. Evidences of a relationship between estrogen and BRCA1 and 2 have been demonstrated in mouse and human models [21]. Recently, *BRCA1* tissue-specificity was investigated through its role in suppressing DNA replication stress. Gene expression related to stress was reported to be higher in *BRCA1* mutation carriers luminal epithelial cells, and more specifically, in estrogen-responsive genes [144]. However, the relationship between *BRCA* genes’ mutation and a predisposition for male cancers remains elusive. There is no clear evidence explaining the predisposition to cancer in male *BRCA2* mutations carriers compared to *BRCA1* mutation carriers. Furthermore, the biological association between *BRCA* genes’ mutation and prostate cancer has yet to be answered.

Similarly, the impact of *BRCA* mutations on fertility remains debated. Whilst studies have revealed an age-related accumulation of DNA DSBs within oocytes and a decrease in the expression of DNA repair mechanism genes [66], the impact of *BRCA* mutations on ovarian ageing has, to-date, been poorly investigated. Only one study based on heterozygous *Brca1*-deficient mice showed a decrease in the pool of PFs at birth and higher rate of γ-H2AX at 4-months old compared to non-mutated mice [68]. In parallel, ovarian ageing was assessed in heterozygous and homozygous *Brca2* transgenic mice that harbored a deletion on exon 27, impairing BRCA2 interaction with RAD51. However, no difference in their PFs number nor in γ-H2AX rate was observed compared to WT mice [68]. Finally, a study on mice deficient for *Brip1* showed also no difference in the number of PFs in *Brip1*-deficient mice compared to WT [45].

While mouse studies do not provide clear conclusions regarding *BRCA* mutations and fertility, human studies are even more challenging. The main difficulties concerning the assessment of the human ovarian reserve are high variability and the use of reliable and robust clinical parameters. In research, the ovarian reserve can be assessed by follicle density counting after tissue staining [145]. However, there are intra- and inter-individual variations that need to be considered [146,147]. Only three studies have reported results based on follicle density, and have demonstrated that ovarian tissue from *BRCA*-mutated patients had a lower follicle density than non-mutated ones [83,104,109]. In clinical practice, the serum AMH level is the most used biomarker to evaluate patients’ ovarian reserve, yet AMH is not a direct indicator of the pool of PFs. Various studies have reported divergent results regarding the AMH level in *BRCA1/2*-mutated and non-mutated carriers diagnosed with BC or healthy (non-) carriers [68,94,95,96,98,99,100,101,102,103,104,105,107,108,148]. AMH levels vary considerably between patients, independently of the population studied, challenging the assessment of the impact of *BRCA* genes on the ovarian reserve. Similarly, the impact of *BRCA1/2* mutations on ovarian response for stimulation and retrieval of oocytes varies from one study to the other [93]. Nevertheless, early fertility counselling including the possibility to cryopreserve oocytes before the occurrence of BC and to perform pre-implantation diagnosis is highly recommended for young *BRCA*-mutation carriers.

Regarding cancer patients, specific guidelines have been published regarding fertility preservation in *BRCA*-mutated patients. Oocyte/embryo cryopreservation is recommended in *BRCA*-mutated patients if the initiation of chemotherapy can be delayed. Due to the fewer number of oocytes expected to be retrieved and limitations related to preimplantation genetic diagnosis (PGD) [124], two consecutive stimulations (Dual-STIM) should be considered for those patients when feasible [149]. The use of aromatase inhibitors, such as letrozole, during the COS treatment has been approved to maintain normal estrogen levels in BC patients [125]. OTC is not the first choice but can be offered as an alternative fertility preservation strategy for young BC patients with *BRCA* mutations when OS is not feasible. The safety of the procedure has been evaluated and livebirths have been achieved successfully. However, due to the increased risk of OC in these patients, it is recommended to remove the ovarian tissue after transplantation [149].

Chemotherapy-induced DNA DSBs is well reported both in mice [150,151,152] and in human oocytes [65,153,154]. *BRCA* mutation carriers are potentially more sensitive to chemotherapies gonadotoxicity compared to non-carriers due to the possible harmful impact on the ovarian reserve of *BRCA* mutations. Therefore, the impact of new therapies developed to cure *BRCA*-mutation patients on the ovarian pool must be further evaluated to provide the optimal counseling regarding fertility preservation in the future.

## 5. Conclusions

Concerns about the impact of *BRCA1/2* mutations on fertility have been raised for more than a decade. Even though the data regarding the evaluation of the ovarian reserve of *BRCA*-mutation (non-) carriers are limited. Due to the high variability in humans, and the unavailability of a consistent parameter to assess PFs pool, studies have reported divergent results and the potential impact of *BRCA* mutations on fertility preservation outcomes remains unclear. However, the accumulation of DNA DSBs has been related to age and is correlated with a diminished expression of DNA repair genes, including *BRCA1* and *BRCA2*. Thus, particular attention should be paid to young *BRCA* mutation carriers, and to individuals exposed to gonadotoxic treatments, as they are more vulnerable to DNA damage and genomic instability. Fertility counselling should be offered to all women diagnosed with a *BRCA* mutation, with or without cancer, as fertility preservation strategies are available for this population.

## Figures and Tables

**Figure 1 ijms-25-00204-f001:**
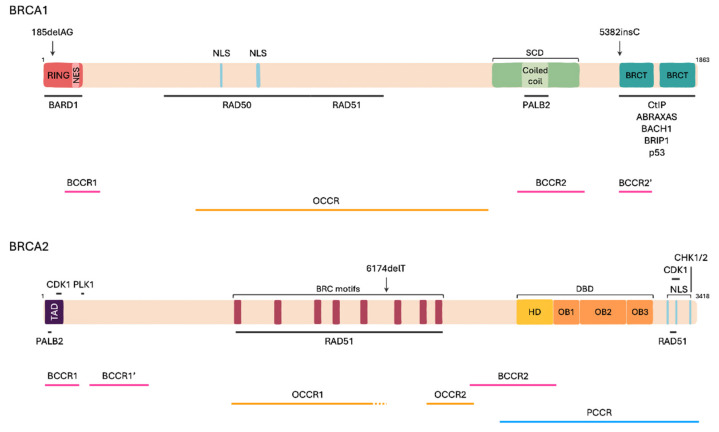
BRCA1 and BRCA2 structure. BReast CAncer 1 (BRCA1) is an 1863 amino acids (aa) protein constituted of a Really Interesting New Gene (RING) domain, a serin cluster domain (SCD) and two BRCA1 C-Terminal (BRCT) domains. Its localization is supported by one nuclear export sequence (NES) and two nuclear localization sequences (NLSs). It interacts with BRCA1-associated RING domain protein 1 (BARD1) through its RING finger motifs, RAD50, RAD51, Partner and Localizer of BRCA2 (PALB2) through a coiled coil domain within SCD and various phosphorylated proteins involved in DNA repair by homologous recombination through its BRCT domains. BRCA2 is a 3418 aa protein composed of a Transcriptional Activation Domain (TAD), eight BRC motifs and a DNA binding domain (DBD) composed of one helical domain (HD) and three oligonucleotide-binding folds (OB1-3). Its nuclear localization is supported by three NLSs. Its phosphorylation is supported by various kinases such as cyclin-dependent kinase 1 (CDK1), polo-like kinase 1 (PLK1) and checkpoint kinases 1 and 2 (CHK1/2). BRCA2 mainly binds to PALB2 and RAD51. Pathogenic founder variants are harbored in both genes (185delAG, 5382insC and 6174delT) and more specifically present in breast cancer cluster regions (BCCR) and ovarian cluster regions (OCCR). BRCA2 also harbors a prostate cancer cluster region (PCCR).

**Figure 2 ijms-25-00204-f002:**
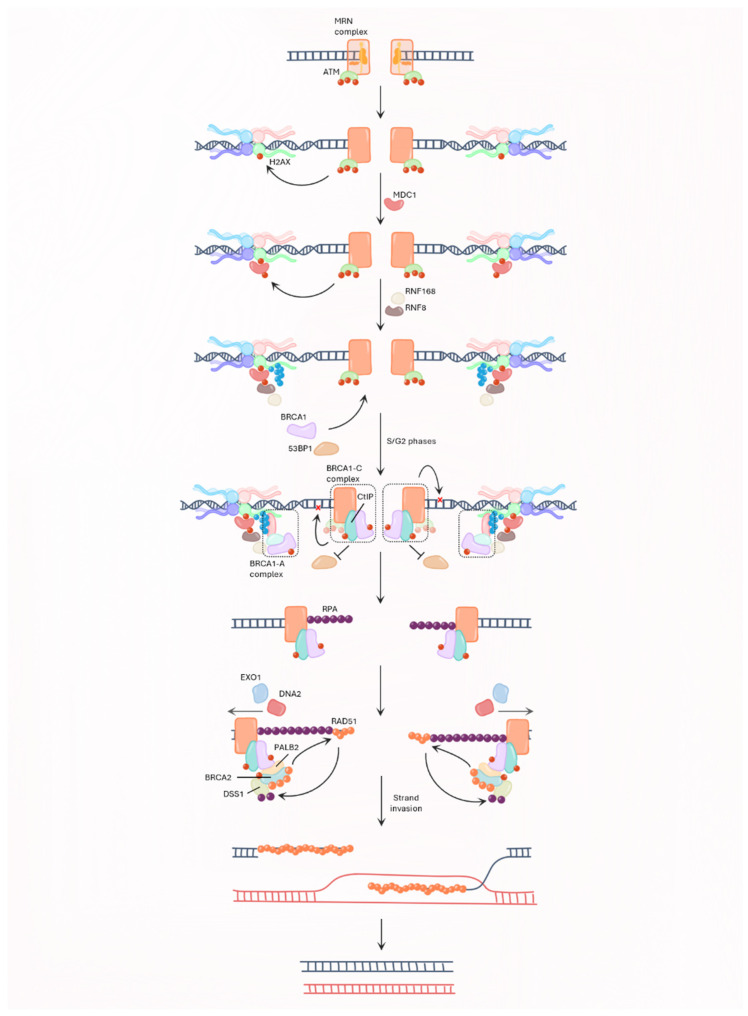
DNA repair through homologous recombination. When DNA double strand breaks (DSBs) occur, the MRN complex (MRE11, RAD50 and NBS1) and ataxia-telangiectasia mutated (ATM) are recruited to the DSB site. ATM then phosphorylates (phosphorylation represented by red circles) H2A histone family member X (γ-H2AX). This signal triggers the recruitment of Mediator of DNA damage Checkpoint Protein 1 (MDC1) and its phosphorylation by ATM. H2AX is then ubiquitylated (ubiquitylation represented by blue circles) by E3-ubiquitin ligases, such as RING Finger proteins 8 and 168 (RNF8 and RNF168), and BReast CAncer 1 (BRCA1) and p53-binding protein 1 (53BP1) are recruited to DSB site. Cells in S/G2 phase form BRCA1-C complex [BRCA1, MRN complex and C-terminal binding protein 1 Interacting Protein (CtIP)] that removes 53BP1 from DSB sites and BRCA1-A complex [BRCA1, ABRAXAS (blue), RAP80 (pink)] that maintains the DNA damage response (DDR). In parallel, BRCA1-C complex initiates end resection with its endonuclease and exonuclease activity (represented by red crosses). The resulting single strand DNA is stabilized by Replication Protein A (RPA). End resection is further supported by exonuclease 1 (EXO1) and DNA2, and RPA is replaced by RAD51 due to BReast CAncer 2 (BRCA2) which is recruited to DSB sites by its interaction with BRCA1 and Partner and Localizer of BRCA2 (PALB2). BRCA2 also interacts with DSS1 to facilitate the switch RPA/RAD51. Strand invasion is the last step of HR, resulting in two homologous chromatids.

**Table 1 ijms-25-00204-t001:** Studies evaluating the ovarian reserve in *BRCA*-mutated patients compared to control patients.

Study	Design	Results
**Anti-mullerian hormone (AMH)**		
Titus et al. (2013) [68]	24 *BRCA*+ patients, 60 control patients	*BRCA*+ vs. control: lower AMH levels (1.22 ± 0.92 ng/mL vs. 2.23 ± 1.56 ng/mL; *p* < 0.0001)
Michaelson-Cohen et al. (2014) [101]	41 *BRCA*+ patients, 324 control patients	*BRCA*+ vs. control: similar AMH levels (2.71 ± 0.59 ng/mL vs. 2.02 ± 0.12 ng/mL; *p* = 0.27)
Phillips et al. (2016) [102]	172 *BRCA1*+ patients vs. 216 control patients for known *BRCA1* mutation and 147 *BRCA2*+ patients vs. 158 control patients for known *BRCA2* mutation	*BRCA1*+ vs. control: 25% lower AMH levels (exp(β) = 0.75; 95% CI = 0.59–0.95; *p* = 0.02) *BRCA2*+ vs. control: similar AMH levels (exp(β) = 0.99; 95% CI = 0.77–1.27; *p* = 0.94)
van Tilborg et al. (2016) [95]	124 *BRCA*+ patients, 131 control patients	*BRCA*+ vs. control: similar AMH levels (1.90 [0.11–19.00] µg/L vs. 1.80 [0.11–10.00] µg/L; *p* = 0.34)
Johnson et al. (2017) [103]	55 *BRCA1*+, 50 *BRCA2*+; 64 control patients	*BRCA1*+ vs. control: similar AMH levels (geometric mean ratio: 1.00; 95% CI 0.7–1.44; *p* = 0.999) *BRCA2*+ vs. control: 33% lower AMH levels (geometric mean ratio: 0.67; 95% CI 0.47–0.94; *p* = 0.037)
Lambertini et al. (2018) [104]	25 *BRCA*+ patients, 60 control patients	*BRCA*+ vs. control: similar AMH levels (1.8 [1.0–2.7] µg/L vs. 2.6 [1.5–4.1] µg/L; *p* = 0.109)
Grynberg et al. (2019) [105]	52 *BRCA*+ patients, 277 control patients	*BRCA*+ vs. control: similar AMH levels (3.6 ± 2.9 ng/mL vs. 4.1 ± 3.6 ng/mL; *p* = 0.3)
Gunnala et al. (2019) [106]	38 *BRCA*+ BC, 53 control BC and 19 *BRCA*+, 600 control	BC-*BRCA*+ vs. control: similar AMH levels (2.6 ± 2.1 ng/mL vs. 2.4 ± 2.4 ng/mL; *p* = 0.915) Cancer-free-*BRCA*+ vs. control: similar AMH levels (3.2 ± 2.2 ng/mL vs. 2.3 ± 2.2 ng/mL; *p* = 0.403)
Son et al. (2019) [107]	52 *BRCA*+ patients, 264 control patients	*BRCA*+ vs. control: lower AMH levels (2.60 ng/mL vs. 3.85 ng/mL; *p* = 0.004)
Ponce et al. (2020) [108]	32 *BRCA1*+ patients, 37 *BRCA2*+, 66 control patients	*BRCA1*+ vs. *BRCA2*+ vs. control: similar AMH levels (3 ± 2.27 ng/mL vs. 2.54 ± 2.07 ng/mL vs. 2.27 ± 2.03 ng/mL; *p* = 0.28) but once adjusted by age showed lower AMH levels in *BRCA2*+ patients (20.2% vs. 23.5% *BRCA1*+ vs. 28.4% control)
Gasparri et al. (2021) [97]	147 *BRCA*+ patients, 405 control patients (age under 42 years)	*BRCA*+ vs. control: lower AMH levels (odds ratio: −0.73 [−1.12, −0.35]; *p* = 0.10]; *p* = 0.0002)
Turan et al. (2021) [96]	246 *BRCA*+ patients, 578 control patients	*BRCA*+ vs. control: lower AMH levels (23% lower; 95% CI, 4 to 38; *p* = 0.02)
Drechsel et al. (2022) [99]	36 *BRCA*+ patients, 126 control patients	*BRCA*+ vs. control: similar AMH levels (2.40 [1.80–3.00] ng/mL vs. 2.15 [1.30–3.40] ng/mL; *p* = 0.45)
Kim et al. (2022) [98]	39 *BRCA*+ patients, 20 control patients	*BRCA*+ vs. control: similar AMH levels (4.2 ± 3.6 ng/mL vs. 5.3 ± 3.5 ng/mL; *p* = 0.173)
El Moujahed et al. (2023) [94]	57 *BRCA*+ patients, 254 control patients	*BRCA*+ vs. control: lower AMH levels (1.6 [0.8–2.9] ng/mL vs. 2.4 [1.4–3.7] ng/mL; *p* = 0.02)
Prokurotaite et al. (2023) [100]	20 *BRCA*+ patients with BC, 10 *BRCA*+ without BC, 55 control patients	*BRCA*+ BC vs. *BRCA*+ without BC vs. control: similar AMH levels (1.7 [0.2–4.7] µg/L vs. 1.8 [0.5–8.3] µg/L vs. 2.3 [0.3–13] µg/L; *p* = 0.22)
**Follicle density**		
Pavone et al. (2014) [109]	35 risk-reducing surgery of which 15 *BRCA*+, 35 control patients (physiological findings)	*BRCA*+ vs. control: lower number of follicles per slide (15.4 vs. 23.3; *p* < 0.05)
Lin et al. (2017) [83]	13 *BRCA1*+ patients, 5 *BRCA2*+ patients, 12 control patients	*BRCA*+ vs. control: lower number of PFs per mm^3^ (11.2 ± 6.7 vs. 44.18 ± 6.1; *p* = 0.0002) *BRCA1*+ and *BRCA2*+ vs. control: *p* = 0.0001 and *p* = 0.0003
Lambertini et al. (2018) [104]	19 *BRCA*+ patients, 53 control patients	*BRCA*+ vs. control: lower number of oocytes per mm^2^ (0.33 vs. 0.78; *p* = 0.153)
**Antral follicle count (AFC)**		
Grynberg et al. (2019) [105]	52 *BRCA*+ patients, 277 control patients	*BRCA*+ vs. control: similar AFC (3.6 ± 2.9 vs. 4.1 ± 3.6; *p* = 0.3)
Gunnala et al. (2019) [106]	38 *BRCA*+ BC, 53 control BC and 19 *BRCA*+ cancer-free, 600 control cancer-free	BC-*BRCA*+ vs. control: similar AFC (15.2 ± 5.0 vs. 13.9 ± 6.3; *p* = 0.757) Cancer-free-*BRCA*+ vs. control: higher AFC (16.3 ± 3.9 vs. 12.2 ± 5.4; *p* = 0.025)
Drechsel et al. (2022) [99]	30 *BRCA*+ patients, 122 control patients	*BRCA*+ vs. control: similar AFC (15.0 [10.8–20.3] vs. 14.5 [9.0–20.0]; *p* = 0.54)

*BRCA*+: BRCA-mutated patients; BC: breast cancer.

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
