# Peer review of "BRCA Mutations and Fertility Preservation"

_ijms, 2023, doi:10.3390/ijms25010204_

Round 1
Reviewer 1 Report
Comments and Suggestions for Authors
This draft paper presents clear and important information about the function of BRCA and the involvement of mutations in the gene in cancer, reproductive organs, and fertility in a clear and accurate manner, citing numerous articles. Thus, I consider this a very valuable review.
However, in Chapter 2, which describes the function of BRCA and the functional changes caused by its mutations, the topic of follicles suddenly comes up (especially in Lines 217-237). Even if the topic is meiosis, this seems abrupt and out of place; perhaps it would be better to move it into Chaper 3.
I hope this comment is helpful.
Author Response
Dear reviewer, thank you for your suggestion. Initially, we wanted to introduce follicles in Chapter 2 to maintain a link with the subject of the review. However, we understand that it may seem sudden and off topic, so we have taken your suggestion into account and moved everything that involves follicles from Chapter 2 to Chapter 3 for smoother reading and better dissociation between BRCA structure/ general function and the role of BRCA in fertility (modifications made at lines 223-243 and 263-283).
Reviewer 2 Report
Comments and Suggestions for Authors
Since mutations in BReast CAncer (BRCA) genes are responsible for the majority cases of hereditary breast and ovarian cancer, as well possibly some male infertility, this manuscript systematically reviewed the progress and questions in this field based on extensive literature analysis. This review can serve as a good reference for the researchers in this field, in the reviewer's opinion. The authors were asked to think the following minor issues.
1. Since section 3.1-3.3 mainly talk about BRCA1/2 with female fertility, so the reviewer think that the Subtitle 3.1 better be changes as "3.1 Ovarian ageing", and "3.3 Fertility preservation" better be changed as " 3.3 Female fertility preservation ". Anyway, the authors need to think about it carefully according to the content.
2. Although most of the manuscript disscused BRCA1/2 with female fertility, it also included a small part 3.4 about BRCA1/2 with male fertility. So are a few words about BRCA with male fertility needed in the Abstract?
Author Response
- Since section 3.1-3.3 mainly talk about BRCA1/2 with female fertility, so the reviewer think that the Subtitle 3.1 better be changes as "3.1 Ovarian ageing", and "3.3 Fertility preservation" better be changed as " 3.3 Female fertility preservation ". Anyway, the authors need to think about it carefully according to the content.
Dear reviewer, thank you for your comment. This review focuses on the role of BRCA mutations in fertility, in both women and men, so it is indeed relevant to specify which model is referred to in the subtitles. Thus, the subtitles have been modified according to your suggested changes (lines 222 and 344).
- Although most of the manuscript disscused BRCA1/2 with female fertility, it also included a small part 3.4 about BRCA1/2 with male fertility. So are a few words about BRCA with male fertility needed in the Abstract?
Thank you for this suggestion. We have modified the abstract to properly introduce the topics covered in Chapter 3 – BRCA mutations and fertility –, including male infertility and a clarification of the fertility preservation model (lines 16-17).